# Non-Systematic Review of Diet and Nutritional Risk Factors of Cardiovascular Disease in Obesity

**DOI:** 10.3390/nu12030814

**Published:** 2020-03-19

**Authors:** Anna Maria Rychter, Alicja Ewa Ratajczak, Agnieszka Zawada, Agnieszka Dobrowolska, Iwona Krela-Kaźmierczak

**Affiliations:** Department of Gastroenterology, Dietetics and Internal Diseases, University of Medical Sciences Poznan, 49 Przybyszewskiego Street, 60-355 Poznan, Poland; alicjaewaratajczak@gmail.com (A.E.R.); aga.zawada@gmail.com (A.Z.); agdob@ump.edu.pl (A.D.)

**Keywords:** cardiovascular disease risk, dietary intake, obesity

## Abstract

Although cardiovascular disease and its risk factors have been widely studied and new methods of diagnosis and treatment have been developed and implemented, the morbidity and mortality levels are still rising—cardiovascular disease is responsible for more than four million deaths each year in Europe alone. Even though nutrition is classified as one of the main and changeable risk factors, the quality of the diet in the majority of people does not follow the recommendations essential for prevention of obesity and cardiovascular disease. It demonstrates the need for better nutritional education in cardiovascular disease prevention and treatment, and the need to emphasize dietary components most relevant in cardiovascular disease. In our non-systematic review, we summarize the most recent knowledge about nutritional risk and prevention in cardiovascular disease and obesity.

## 1. Introduction

Cardiovascular disease (CVD) is a leading cause of death worldwide and its prevalence is still increasing [1,2,3]. CVD is responsible for more than four million deaths each year in Europe and almost two million deaths in the European Union, accounting for 45% and 37% of total deaths respectively [4,5]. The number of deaths from CVD is higher in women than men, with CVD accounting for 49% and 40% of all deaths in women and men respectively [1,4]. Over half of the global adult population has excessive body weight, according to World Health Organization (WHO) statistics [5,6,7]. Numerous clinical and epidemiological data have linked obesity to a comprehensive spectrum of CVD including heart diseases, arrhythmias, hypertension, and cardiac death [5]. Obesity is also associated with obstructive sleep apnea and other hypoventilation syndromes, which negatively affect cardiovascular health [5,6,7,8]. Nutrition is one of the major factors influencing cardiovascular risk directly by physiological, molecular, and biological changes with inflammatory initiation and oxidative stress [9,10]. It may also impact on CVD development indirectly by affecting body mass, blood pressure, lipid profile, and the risk of atherosclerosis and diabetes [11,12,13,14,15]. Although nutrition is a modifiable risk factor, many people, especially those with obesity, do not follow the dietary recommendations for cardiovascular disease prevention [11,16,17]. Most adults include more protein, fat, saturated fatty acids, and cholesterol in their diet than it is recommended [11,18]. Furthermore, vitamin (especially A, C, and B_1_), micronutrient (potassium, calcium, magnesium, and iron), and dietary fiber intake is insufficient [11,18]. It shows the need for ongoing implementation of prevention, especially in the field of proper nutrition, and dietary habits. In our non-systematic review, we want to summarize the most recent and crucial nutritional aspects of cardiovascular disease prevention.

## 2. Obesity and Its Association with CVD

According to the American Heart Association and the World Health Organization, obesity is defined as a major modifiable risk factor for CVD [5,19,20]. Epidemiological studies have shown a relationship between obesity and cardiovascular morbidity and CVD mortality [19,21,22,23,24,25,26]. WHO emphasizes that obesity is strongly related to major cardiovascular risk factors, such as raised blood pressure, glucose intolerance, type 2 diabetes, and dyslipidemia [19,21,24,25,27]. Weight loss programs using diets, physical activity, or behavioral interventions have been recommended [19].

The relationship between obesity and CVD may be partially determined by obesity-related comorbidities, hemodynamic repercussions of obesity, body fat mass content, and distribution. Obesity is usually defined, in clinical practice, by body mass index (BMI) [13]. WHO defines obesity as excess body fat affecting negatively on health [7]. However, it is suggested that BMI may not be a good predictor of body composition, simply because different subjects with a similar body mass index may present diverse levels of body fat content [7]. Classes of BMI and cardiovascular disease mortality may differ in older adults compared to middle-aged adults. For older populations, BMI <23.0 may be associated with increased all-cause mortality when BMI range 18.5–24.99 is considered appropriate in middle-aged adults [28]. Due to mentioned BMI limitations, meta-analysis by Flegal et al. led to very controversial results. The authors analyzed that relative to normal weight, overweight, and mild obesity (class I) subjects, had lower or consistent CVD mortality respectively [29]. It could be suggested that more accurate measures of body fat mass could be better predictors of mortality. There has been, however, a proposal from the European Association of the Study of Obesity (EASO) to improve the diagnostic criteria for obesity [30]. The authors suggested changing the structure of classification of obesity according to three-dimensional classification: obesity etiology (dimension 1), degree of adiposity (dimension 2), and health risk (dimension 3). The summary of those proposed changes in adults is shown in the table below (Table 1). Even though there are several limitations of BMI in providing information about body composition and adiposity, it is often the best available measure [13]. BMI was a more valuable predictor of CV (cardiovascular) mortality than indicators of fat mass in over 60 thousand participants in Ortega et al. study [31].

The adipose tissue volume is unquestionably linked to cardiovascular risk, but recent evidence indicates that fat tissue distribution, so-called ’fat quality’, also matters [14,32,33]. Abnormalities in adipose tissue may be factors that regulate systemic metabolism and lead to cardiometabolic disease, independent of BMI [3]. It is now identified that the development of visceral adipose tissue (VAT) is heavily associated with elevated CV risk when an increase of subcutaneous adipose tissue (SAT) has a minor contribution on metabolic dysfunction [3]. The adipose tissue is not only a passive repository for fat but is also an endocrine organ, capable of synthesizing and releasing an essential diversity of peptides and nonpeptide compounds—adipokines, interleukin 6, C-reactive protein (CRP), and tumor necrosis factor (TNF-α) [8,32,34]. These compounds may have a function in maintaining CV homeostasis [31]. Visceral fat (not subcutaneous) volume may be more strongly associated with systemic endothelial dysfunction [3,12]. Although, Abraham et al. observed subcutaneous fat in women may have negative metabolic effects apart from total adiposity. It may be clarified by the evidence that women usually have higher levels of SAT than men [15,32]. It may also be observed because in women, SAT is less likely to proliferate new cells and expand adipose cells [12,32]. However, the authors, after further analysis, noticed that the influence of SAT on cardiovascular risk in women was not different than VAT volume.

Sarcopenia is the age-related loss of muscle mass with decreased strength and function, with several negative outcomes such as physical disability and poor quality of life [35]. Another clinical condition linked to sarcopenia is sarcopenic obesity. It is characterized by the coexistence of excessive adipose tissue and sarcopenia (low skeletal muscle mass) [36]. The pathogenesis of this clinical condition is dependent on a variety of factors, such as sedentary lifestyle, unhealthy diet, systemic inflammation, and oxidative stress [37]. Sarcopenic obesity mostly occurs in older subjects but may also be diagnosed within the younger obese subjects with several acute or chronic diseases. According to the recent knowledge, early diagnosis of sarcopenic obesity is important in prevention and therapeutic strategies in cardiovascular disease [38]. Markers of oxidative stress, determinant in atherosclerosis, are increased in patients with sarcopenic obesity, and are essential in measuring cardiovascular risk [38]. Serum hs-CRP and sICAM-1 were negatively associated with muscle strength and positively associated with body fat percentage in peritoneal dialysis patients [39]. Additionally, sarcopenic obesity may be associated with elevated triglycerides, total cholesterol, and decreases in high-density lipoprotein [40].

Obesity, influencing hemodynamic adjustment, may provoke alteration in cardiac anatomy which affects left and right ventricular dysfunction and heart failure, even if other manifestations of heart disease are not present [18,41]. The primary cardiac morphology changes due to obesity are: an increase in total circulating blood volume, increased cardiac output, and increased systemic vascular resistance [5,6]. The term obesity-associated cardiomyopathy attributes to the gradual replacement of the myocardium by non-regular bands of adipose tissue, which separate and cause pressure-induced atrophy of myocardial cells. The clinical spectrum may range from subclinical LV alterations without any present symptoms to overt dilated cardiomyopathy [5,6,42].

Other types of adipose tissue relevant in cardiovascular disease are epicardial adipose tissue (EAT), and pericardial adipose tissue (PAT). EAT is located between the myocardium and visceral layer of the pericardium and interacts directly with the myocardium due to its proximity [43]. It acts mainly locally and produces a variety of bioactive molecules (e.g., proinflammatory cytokines; profibrotic factors; adipokines; and chemokines) that may have both harmful and protective effects on cardiac function and morphology [44,45]. Environmental factors including excessive caloric intake and an unhealthy lifestyle may drive the development of ectopic fat in the heart [44]. EAT affects cardiac function by increasing the left ventricular chamber (LV) mass and atypical right ventricular chamber (RV) geometry [44]. Due to the profibrotic, proinflammatory, and oxidative stress mechanisms it is also associated with severity and prevalence of the most prevalent cardiac arrythmia—atrial fibrillation (AF) [4,46]. Epicardial adipose tissue volume is related to not only the presence but also the severity of coronary artery disease (CAD), as the Kim et al. study has shown [47]. It is also associated with subclinical atherosclerosis, however, its effects may be mediated by hyperglycemia or dyslipidemia [48,49]. It also plays a role in the initiation of heart failure [50]. PAT surrounds the heart and directly contacts with the coronary arteries [51]. Similarly to EAT, PAT also correlates with atherosclerosis, atrial fibrillation, coronary plaque, and coronary artery disease [52,53,54,55,56]. It also influences LV function, however a large-scale study on participants from the Framingham Heart Study reported that this association may not be stronger than it is with the visceral adiposity [52,57]. 

## 3. Obesity Paradox

It is commonly known that obesity is one of the main causes of numerous comorbidities, such as cardiovascular disease, type 2 diabetes (T2D), dyslipidemia, and hypertension [6,58]. Evidence has demonstrated that patients with excessive body mass may be in favor of better CVD prognosis when compared with normal BMI patients. This discovery has been named the “obesity paradox” [59,60]. This term refers to patients that despite having excessive body fat have a better prognosis compared with leaner patients [61]. Obesity paradox may be related to the term MHO – Metabolically Healthy Obesity [62]. MHO/obesity paradox is associated with better insulin sensitivity, immune profile, lipid profile, and no hypertension [62]. This condition may refer to 10%–30% of obese patients [63]. Although several possible explanations have been suggested, the debate on the obesity paradox is still ongoing. 

In the Uretsky et al. study on over 22,000 patients with hypertension, all-cause mortality was one third lower in overweight and obese hypertensive patients compared with normal weight hypertensive patients [64]. Coronary heart disease (CHD) patients with excessive body weight have a smaller risk of cardiovascular and overall mortality in comparison with underweight and normal-weight patients, as Romero-Corral et al. meta-analysis reported [65]. However, in patients with a BMI classified as class II obesity, risk of cardiovascular mortality was increased [65]. Similar findings have been observed in patients with heart failure and atrial fibrillation [59,66,67,68]. Although the obesity paradox can be confirmed in studies, it is necessary to keep in mind that morbid (III class) obesity is one of the dominant risk factors for the development of cardiovascular diseases and is associated with worse prognosis when CVD symptoms become present [68]. Also, the obesity paradox may be associated with developing CVD in a longer follow-up period [63]. It is essential to remember that the obesity paradox may be explained by limited understanding of the complex pathophysiology of obesity and its association with CVD [10]. As was mentioned before, BMI accuracy of obesity diagnosis may be questioned which can affect the results of studies investigating the obesity paradox. It may also be explained by several factors that should be considered. CVD patients are usually older and have more comorbidities, while obese CVD patients are usually younger and receive earlier and more intensive care [69,70]. Furthermore, older CVD patient usually have lower BMI values [70]. Elderly populations with lower BMI values may still have an excessive visceral and overall adiposity, which even in, healthier individuals (i.e., nonsmokers) may lead to higher CVD mortality [70]. Aging also negatively affects the CV system, including cardiac remodeling and hemodynamic function. There are also other confounding factors like type of obesity, CVD duration or severity, age, smoking, physical activity, and cardiorespiratory fitness that are heavily related to CVD prediction and may influence the relationship between body weight and mortality [59,71]. Furthermore, greater lean mass, which can be reflected in a higher BMI, is associated with better cardiorespiratory fitness levels. Greater fitness levels may be associated with an improved health condition [72,73]. 

## 4. Nutrition-Related CV Risk Factors 

As was mentioned before, CVD is the leading worldwide cause of death in Western nations [74]. Atherosclerosis is an inflammatory disease that leads to a major prevalence and mortality of cardiovascular disease. The inflammatory pathogenesis and risk factors of atherosclerosis are fully characterized. Many markers of inflammation such as high sensitivity CRP (hs-CRP), IL (interleukin)-6, and fibrinogen have been associated with an increased risk of CV events, independent of the cholesterol level [75,76,77]. Activated immune cells in the plaque produce many inflammatory cytokines, such as interferon, IL-1, and TNF-α, which induce the production of IL-6 [78]. This is evidenced by the Canakinumab Anti-inflammatory Thrombosis Outcome Study (CANTOS), the first one to directly test the inflammatory hypothesis of atherosclerosis. Canakinumab, a fully human monoclonal antibody that neutralizes IL-1β, significantly reduced the rate of recurrent CV events in patients with prior myocardial infarction in the CANTOS study [78,79].

Nonetheless, numerous health statuses and habits may lead to atherosclerosis development, such as low high-density lipoprotein (HDL-C) and high total-cholesterol (TC) levels, hypertension, T2D, excessive body-weight, and physical inactivity [80]. Furthermore, beneficial dietary patterns and lifestyle corrections are conceivable approaches for atherosclerosis and oxidative stress prevention. The possibility of developing CVD is associated with unhealthy dietary patterns (i.e., disproportionate intake of refined foods and sodium; added sugars; unhealthy fats; and a low intake of fruit, whole grains, fiber, legumes, fish, vegetables and nuts) together with low levels of physical activity, excessive body weight, alcohol consumption, stress, or smoking [6,32,67,81]. A high-quality diet may lower the risk of developing chronic disease by improving lipid and glucose metabolism, lowering chronic inflammation, and increasing levels of lipid-soluble micronutrients [32,82]. Daily consumption of fish, tea, and vegetable oils were significantly associated with a lower risk of coronary events, Amani et al. revealed [83]. Consumption of yogurt has neutral or favorable influence on CVD risk [84,85]. Green leafy vegetables and those rich in carotenoids and vitamin C may partly reduce cardiovascular risk [80,83]. Another essential part of nutrition affecting CVD are micronutrients. Micronutrients apply their protective impact by a number of possible mechanisms including: inhibiting oxidation of LDL-C (low-density lipoprotein), improving the production of NO, and reducing endothelial cells damage [41,81]. Pro-inflammatory biomarkers have been correlated with Se, Zn, and vitamin E and C levels [34,83,86,87]. Deficiency of those micronutrients may indicate a higher CVD risk. 

## 5. Appropriate Diet in CVD Prevention

Nutritional studies have focused on the role of essential nutrients to prevent deficiencies, but at the present time, the nutritional strategies are essential in promoting health and reducing noncommunicable disease prevalence [8]. In numerous states, diet is a major active force, which is often easier to change and follow than other factors [11]. Nevertheless, as a big study on the Polish population (WOBASZ II) concluded, the majority of adults with obesity fall far short of the recommendations relevant to CVD prevention [11]. Although, a lot of the participants included in the WOBASZ II study considered their diet to be appropriate. One of the key strategies to treat cardiovascular disease is to adjust lifestyle habits and focus on the beneficial properties of exact nutrients [8,88]. The dietary risk seems to be a priority target for cardiovascular disease prevention and treatment [59,89].

### 5.1. The Mediterranean Diet

To one of the specific diets, or as we should call it, a collection of eating habits, we can include the Mediterranean diet (MeD) [90,91]. The MeD diet is a plant-centered nutritional approach with a high intake of vegetables, fruits, whole grain cereals, and legumes [90,91,92,93,94]. It is also characterized by abundant use of olive oil and moderate consumption of wine, especially red [92]. Fish and poultry consumptionshould be moderate, as well as dairy products. A characteristic of MeD is a low consumption of red meat and sweets [9,60,92]. More detailed dietary recommendations are listed below (Table 2). It should be noticed that MeD has variants, which depend on the characteristics of Mediterranean Sea population, thus makes it challenging to establish a unique universal definition [91]. Evidence suggests that MeD is negatively associated with the risk factors of metabolic syndrome and diabetes [92,93,94]. It also lowers low-density lipoprotein and triglycerides levels, reduces body weight, and has a beneficial effect on blood pressure [91]. The PREDIMED (Prevención con Dieta Mediterránea) study was carried out in Spain for the primary prevention of cardiovascular disease with MeD [93]. The investigators have focused on the influence of MeD enriched with a high olive oil or nut intake. It has shown that a diet rich in high-unsaturated fat is more favorable than a low-fat diet. Dietary patterns used in the PREDIMED showed a reduction of CVD risk by approximately 30% [92,94,95]. Participants included in the study were 55–75 years old (men) or 60–80 years old (women). It shows that even late dietary changes may be beneficial. However, there have been some controversies concerning the PREDIMED trial associated with the study design. The PREDIMED-PLUS study has been conducted and has focused on the promotion of physical activity and weight-loss goals which may reduce CVD risk even more. The results of this study, however, have not been published yet. It should be pointed out that even though an MeD dietary pattern is valuable in cardiovascular disease prevention, future randomized-controlled trials should be conducted to demonstrate MeD effects in other non-Mediterranean populations to determine the transferability of the MeD recommendations [60,91,96].

### 5.2. The DASH Diet

Dietary approaches to stop hypertension (DASH) is a dietary pattern, which focuses on fruit, vegetables, fat-free/low-fat dairy, whole grains, nuts, and legumes intake, which are the favorable food groups [6,97]. Saturated fat, cholesterol, red and processed meats, sweets, sodium, added sugars, and sugar-sweetened beverages should be limited since they belong to the unfavorable food group [97]. More detailed dietary recommendations are listed below (Table 3). The most significant difference between the MeD and the DASH diet is the use of extra-virgin olive oil which is more prominent in the first diet [12]. A Schwingshackl et al. study on the consumption of a DASH diet found that it reduced CVD prevalence [97,98]. Additionally, the DASH diet was found to significantly lower systolic and diastolic blood pressure, TC, and LDL-C with no significant effect on HDL-C or triglycerides (TG) [99]. Although the DASH dietary pattern is a well-accepted nutritional approach associated with CVD benefits, new studies should be conducted in order to estimate and confirm its positive effect in clinical practice [6,99].

### 5.3. Plant-Based Diets

Plant-based diets have been progressively favored for their health benefits. Vegetarian diets can exclude the intake of a few or all animal foods. Vegan diets eliminate the consumption of all animal products. Lacto-vegetarians consume only dairy products and lacto-ovo-vegetarians consume eggs and dairy products [2,101]. Some studies have also investigated semi-vegetarian diets, which are occasionally defined in terms of exclusion of just red meat, and at other times as infrequent intake of poultry and red meat [2,101]. The latter of these vegetarian dietary patterns is consistent with dietary guidelines, with the World Cancer Research Fund suggesting limiting red and processed meat to 350–500g per week in their cancer-prevention recommendations [102]. Despite the broad variety of plant-based diets in the evidence, their associations with CVD in prospective cohort studies have been fairly homogenous [101]. Key et al. found an over 20% lower rate of CHD mortality among vegetarians relative to omnivores, but no association was found with stroke mortality [90,101]. These findings were consistent with two different, more recent meta-analyses [103,104]. In most of the studies, the inverse associations were substantial among younger subjects, among subjects with a longer period of attachment to a vegetarian diet, and among men relative to women [101]. Additionally, a lower rate of stroke mortality among vegetarian men relative to omnivore men was found, but the association was not significant among women [104]. Wang et al. found in their meta-analysis of randomized-controlled trials that vegetarian diets are significantly lower in LDL-C, HDL-C, and total cholesterol relative to a range of omnivorous control diets [105]. Vegetarian diets lower blood pressure to a larger extent than omnivorous diets, Yokoama et al. found [106]. The favorable effects of the plant-based diet on traditional risk factors of CVD found in randomized-controlled studies, and their contrary associations with hard cardiovascular endpoints found in prospective cohort studies present strong support for the approval of healthy plant-based diets for CVD prevention [101]. This type of diet is more likely to be low in energy density, which could help with favorable weight loss and its maintenance. For individuals who prefer not to consume particular or most animal foods, the healthy and well-planned plant-based dietary pattern can contribute sufficient nutrition and CV health benefits [2,101].

### 5.4. The Portfolio Dietary Pattern

The portfolio dietary pattern is focused on four cholesterol-lowering foods [6,107]. Every cholesterol-lowering food has a Food and Drug Administration, Health Canada, and/or a European Food Safety Authority approved health claim for cardiovascular disease risk [6,107]. These products are nuts, apples, oranges, berries, soluble fiber from oats, barley, psyllium, okra or eggplant, plant protein from dietary pulses or soy products, and 2g plant sterols provided in a plant sterol-enriched margarine [6,107,108]. This dietary pattern results in clinically meaningful reductions in the primary therapeutic lipid target for CVD prevention as well as blood pressure, CRP level, culminating in an improvement in estimated 10-year CHD risk [6,107]

### 5.5. Low-Carbohydrate Diet

Low-carbohydrate diet interest has been increasing for reducing body weight as well as weight management [4]. Nonetheless, it is not often recommended in the main guidelines because of the fear of its adverse effects on CVD risk, mainly due the belief that saturated fats intake will be increased when the carbohydrate will be limited [4,109,110]. Hu et al. in their searched MEDLINE online database to identify studies (lasted at least 6 months) that examined the low-carbohydrate diet (≤45% of energy from carbohydrates, also ketogenic diet) compared to the low-fat diet (≤30% of energy from fat) in adults [109]. Both low-carbohydrate and low-fat diets appeared to improve the lipids profiles (total cholesterol, HDL-C, LDL-C, total/HDL ratio, and triglycerides), without robust evidence that either one was better. Available randomized-controlled trials determined that low-carbohydrate diets decrease blood pressure to a similar level as isocaloric low-fat diets [96,111,112,113]. The low-carbohydrate diets may also improve inflammatory status, carotid endothelial function, homocysteine, and adipokines levels [114,115,116]. It should be pointed out that these data come from small clinical trials and require further extensive research. The low-carbohydrate diets had positive effects on weight loss and cardiovascular risk factors, comparable to those seen on low-fat diets. Additionally, ketogenic diets did not show a higher reduction in body weight or a better improvement in cardiovascular risk factors than non-ketogenic low-carbohydrate diets. It is worth noticing that ketogenic diets are challenging to follow long-term, and non-ketogenic low-carbohydrate diets may be more efficient for obtaining weight loss and improving CV risk [111,113,114]. The Paleolithic diet is another type of low-carbohydrate diet. It is characterized by a high intake of lean meat, vegetables, fruits, and nuts and a very small intake of grains, dairy, sugar, and salt. Some studies conducted say that a Paleolithic diet may improve metabolic syndrome, however, changes in primary outcomes (lipid profile, blood pressure, fasting blood glucose) were small [117]. The Paleolithic diet may lead to deficiencies of some micronutrients, e.g., calcium. Additionally, it is often high in cholesterol, protein and saturated fatty acids which may have an adverse effect on CVD [118]. It could be high in *n*-3 PUFA due to a high consumption of fish, and could reduce TG levels [118]. However, not all Paleolithic diets are rich in *n*-3, and these results should not be used as an overall influence. Trimethylamine-N-oxide (TMAO) is produced by microbiota due to the consumption of milk, eggs, and meat [119]. Some studies have suggested that TMAO may be a predictor of CVD risk, since patients with heart failure had higher plasma TMAO levels [120]. However, in the Genoni et al. study, no difference in TMAO levels was observed between the Paleolithic diet and the Australian Guide to Healthy Eating diet [119]. 

### 5.6. Body Weight Management

According to The European Atherosclerosis Society (EAS) and The European Society of Cardiology (ESC) recommendations, obesity, especially abdominal adiposity, often contributes to dyslipidemia [12,82]. To obtain weight loss and reduce abdominal adiposity caloric intake should be lowered and energy expenditure increased [12]. Even modest (5%–10%) body weight reduction corrects the lipid profile and has a favorable effect on CV risk factors overall [12]. Even so, the benefits of weight reduction on mortality and cardiovascular disease outcome are less obvious [12]. Although, Nordstoga et al. in a recent study demonstrated that overall and cardiovascular mortality is substantially higher among people who remained inactive and gained weight over a 10 year period [121]. In comparison to physically active (during leisure time) subjects with a stable weight, physically inactive participants who gained weight had a extensively greater cardiovascular and all-cause mortality [121]. Body weight reduction also affects LDL-C and TC, but the capacity of the effect is modest: in obese people, a reduction in LDL-C concentration of 0.2 mmol/L per every 10 kg of weight less is noted for every 10 kg of weight reduction [12]. Weight loss increases HDL-C levels; a 0.01 mmol/L increase is noticed for every kilogram reduction when weight reduction has been maintained [12].

## 6. Diet, Nutrients, and Microbiota in Cardiovascular Diseases

The intestinal microbiota constitutes a very complex ecosystem. It performs many functions, such as maintaining homeostasis of the body, synthesis of vitamins, regulation of absorption of micro- and macro elements, neutralizing toxins, fermentation of undigested foods, and the production of short-chain fatty acids which are important for the body [122]. In obese persons, however, there is a decrease in the number of bacteria from the *Bacteroides* phylum in favor of the growth of the *Firmicutes* [123,124]. Improper diet generates the onset of intestinal dysbiosis and contributes to the development of obesity and diabetes—the main components of the metabolic syndrome—thus increasing cardiovascular risk [125]. The beneficial effect of the vegetarian diet on reducing the risk of cardiovascular disease is particularly associated with a low intake of saturated fatty acids, cholesterol, sodium, and omega 3 fatty acids, together with the increased intake of fiber, folic acid, magnesium, vitamin C, and omega-6 fatty acids [126]. With the consumption of trimethylamine-containing meat products, such as phosphatidylcholine, choline, and L-carnitine, the microbiota produces trimethylamine N-oxide (TMAO). This metabolite together with increased levels of L-carnitine, choline, and phosphatidylcholine are direct cardiovascular risk factors [127,128,129]. In addition, other nutrients in the diet can modulate intestinal microbes for cardiovascular disease. Polyphenols have a positive effect on the diversity of intestinal microorganisms and their increased supply corrects the unfavorable *Firmicutes* to *Bacteroidetes* ratio (F/B ratio) [130]. Vitamins of B and K groups (mainly synthesized by microbiota) affect the synthesis of blood coagulation factors, which also significantly affects the proper functioning of the circulatory system [131]. Vitamin C as the main oxidant and free radical removal factor significantly increases the number of Lactobacillus and Bifidobacterium and reduces the amount of *Escherichia coli* in the intestinal environment [132]. Furthermore, the study of Zuoi et al. showed that vitamin D3 levels were positively correlated with beneficial bacteria: *Subdoligranulum, Ruminiclostridium, Intestinimonas, Pseudoflavonifractor, Paenibacillus,* and *Marvinbryantia*. In addition, these bacteria were claimed to be antihypertensive [133].

### 6.1. Microbiota and Hypertension 

Changes in the intestinal microbiota are also observed in patients with hypertension. This is associated with imbalance of *Bacterioides* and *Firmicutes* as well as with reduced short-chain fatty acid (SCFA) production in this group of people [134,135]. In addition, SCFA can stimulate GPR41 and GPR43 protein-coupled receptors that are found in the renal vessels, and the Olf78 receptor in the kidneys which also regulates blood pressure [136,137]. Experimental studies have also found that rats treated with angiotensin II showed reduced diversity of intestinal biota species and increased the *Firmicutes* to *Bacteroidetes* ratio when compared to control rats [138]. In addition, in the study by Kaye et al., the lack of prebiotic dietary fiber and SCFA produced in the intestine predisposed C57BL/6J mice to the development of hypertension and pathological remodeling of the heart muscle. Transfer of hypertensinogenic microbiota to gnotobiotic mice caused hypertension. Reintroduction of SCFA in fiber-depleted mice had a protective effect against hypertension, hypertrophy, and myocardial fibrosis [139].

### 6.2. Heart Failure and Microbiota

In heart failure, the intestinal barrier permeability increases due to two mechanisms. At the initial stage, owing to reduced cardiac output, blood flow to the intestinal endothelium decreases and intestinal epithelial barrier permeability is increased through ischemia of the intestinal wall [140]. At the advanced stage of heart failure, there is congestion and swelling of the intestinal wall which also increases intestinal permeability. This is directly related to the translocation of endotoxins, microbial components and microbial metabolites such as lipopolysaccharides (LPS) into the systemic circulation [141,142]. As a result, pro-inflammatory cytokines are activated and higher levels of inflammatory markers (C-reactive protein and interleukin-6) are observed in plasma [143].

### 6.3. Microbiota and Diabetes

The gut microbiota of diabetics is different when compared with non-diabetics. Patients with T2D have a low number of butyrate-producing bacteria, and a high number of opportunistic pathogenic bacteria [144]. Butyrate-producing bacteria seem to be associated with glycemia regulation and improved insulin sensitivity [145,146]. Butyrate is significant in the maintenance of intestinal cell integrity, which is essential in the prevention of “leaky gut” observed within diabetic patients [146]. Additionally, *Roseburia* spp., butyrate-producing bacteria, may stimulate defense against oxidative stress [144]. Recent data have also suggested the beneficial effect of metformin, a drug used in T2D, on gut microbiota, especially *Akkermansia* spp. [147,148]

## 7. Current Dietary Recommendations

Lifestyle modifications and nutrition in the prevention of cardiovascular disease were strongly highlighted in the previously-mentioned ESC/EAS recommendations and WHO guidelines for the assessment and management of cardiovascular risk [12,19,20]. A Summary of the most important knowledge is shown in the table below (Table 4). 

## 8. Conclusions

Cardiovascular disease is a leading cause of death worldwide and continues to increase in prevalence. Since nutrition is one of the most crucial and modifiable risk factors, there is a need for better nutritional education of patients with or at risk of CVD. It is essential to achieve normal body weight, considering obesity as a risk factor. Even though studies confirmed that the obesity paradox is present in some obese patients, it is necessary to have in mind that morbid obesity is associated with poor prognosis of the disease development. Unhealthy dietary patterns (i.e., a disproportionate intake of refined foods and sodium, added sugars, and unhealthy fats and a low intake of fruit, whole grains, fiber, legumes, fish, vegetables, and nuts) together with physical inactivity, alcohol consumption, stress, or a smoking habit are increasing the probability of developing CVD. The Mediterranean and DASH diet are recommended dietary patterns to follow to reduce CVD risk, however there is a growing interest in other dietary patterns (e.g., plant-based diets and portfolio diets) that may be favorable for CVD risk management. 

## Figures and Tables

**Table 1 nutrients-12-00814-t001:** Changes in the structure of the classification of obesity according to the European Association of the Study of Obesity (EASO), based on Heberand et. al, 2017.

Changes in Accordance with:			
Etiology (Dimension 1)	Degree of Adiposity (dimension 2)	Health Risk (Dimension 3)	
Obesity, multifactorial	Obesity class I	Low	Absence of risk and obesity/adiposity related diseases
	BMI range: 30.0–34.9 kg/m^2^ Medical risk: associated with obesity, slightly elevated *		
Obesity, attributable to a certain defined etiological factor (arising from, or aggravated by, or due to): iatrogenic drug-induced weight gain other iatrogenic procedures certain defined disease/condition certain defined endocrine disease: - certain defined neoplasm major depressive disorder immobilization/inactivity more than one etiological factor monogenic disease other major disease	Obesity class II	Inter-mediate	Positive family history for adiposity related diseases Visceral adiposity Obesity - class ≥ 1 (age ≤30), - class ≥ 2 (age >30) IGT HT HCL or HTG regular tobacco use physical inactivity
	BMI range: 35.0–39.9 kg/m^2^ Medical risk: associated with obesity, elevated *		
	Obesity class III		
	BMI range: 40.0–44.9 kg/m^2^		
	Obesity class IV		
	BMI range: 45.0–49.9 kg/m^2^		
	Obesity class V		
	BMI range: 50.0–54.9 kg/m^2^		
	Obesity class VI	High	Presence of: T2D, MetS, CV/renal organ damage, OR musculoskeletal disorders
	BMI range: >55,0 kg/m^2^		
	Medical risk (class III-VI): associated with obesity, substantially elevated *		

T2D: type 2 diabetes, MetS: metabolic syndrome, BMI: body mass index, CV: cardiovascular, OR: obesity related, IGT: impaired glucose tolerance, HT: hypertension, HCL: hypercholesterolemia, HTG: hypertriglyceridemia, *: in comparison to normal weight.

**Table 2 nutrients-12-00814-t002:** Dietary recommendations according to the Mediterranean diet.

Food Group	Recommendation	Reference
Nuts	≥3 servings/day (around 30 g)	[9,60,92]
Olive oil	≥4 tbps/day (around 50 mL)	[9,60,92]
Fresh fruits and vegetables	≥2–3 servings/day	[60,93]
Legumes	≥3 serving/day	[60,93]
Fish, poultry, dairy products	≥3 servings/day	[60,93]
Whole grain cereals	75–90 g/day	[9,60]
Wine (red, dry)	≥7 glasses/week	[9,60,93]
Red and processed meats, sweets	<1 serving/day	[92,93,94]

**Table 3 nutrients-12-00814-t003:** Dietary recommendations according to the Dietary approaches to stop hypertension (DASH) diet.

Dietary Product/Nutrient	Recommendation	Reference
Oilseed, seed	4–5 servings/week	[6,97,99,100]
Whole cereals	7–8 servings/day	
Dairy products	2–3 servings/day, low or no fat	
Fruit, vegetables	4–5 servings/day each	
Oil and fats (vegetable)	2–3 servings/day	
Red and processed meats, poultry	≤2 servings/day	
Sweets, added sugars	<5 servings/week	

**Table 4 nutrients-12-00814-t004:** ESC/EAS and WHO nutritional and behavioral recommendations in CVD.

Variable	Recommendation	Reference
Trans-fat	To avoid	[12,149,150]
Saturated fat	- <10% *, - <7% * when hypercholesterolaemia is present	[12,19,149,150]
Dietary cholesterol	- <300 mg/day (especially when plasma cholesterol levels are elevated) - notice there is an individual variation of how dietary cholesterol may influence serum cholesterol	[19,151]
Total fat intake	- Large range of total fat intake; <30% according to WHO guidelines - Fat intake >35% * is not recommended - Not too low (due to possible vitamin E deficiency which may advance to a reduction of HDL-C) - Mainly from sources of monosaturated fatty acids (PUFAs and *n*-6)	[12,19,152]
Carbohydrates	- “Neutral” effect on LDL-C - Excessive intake is not recommended (due to its effect untoward plasma HDL-C and TGs levels) - Total intake around 45–55%* - Added sugars <10% *	[12,19,149,153]
Dietary fiber	- Between 25–40 g per day - Hypocholesterolaemic effect	[12,154]
Fruit and vegetables intake	- At least 400 g - mostly raw and cooked	[12,19,20]
Dietary sodium	- <5 g (90 mmol)/day	[12,19,20]
Phytosterols	- 2 g/day may productively decrease LDL-C (8–10%) and TC (6–9%) levels, no effect on TG and HDL-C levels - Using under consideration of individual indications	[12,155]
Soft drinks	- Limited - Highly restrained TG values are elevated	[12,154
Alcohol	- Moderate consumption acceptable if TG levels are not elevated	[12,154]
Body weight	- BMI 20–25 kg/m2, and waist circumference <94 cm (men) and <80 cm (women) - even modest weight loss of 5–10% is recommended	[12]
Physical activity	- at least 30–60 min of moderate physical activity/day	[12,19,121]
Smoking	- smoking cessation recommended - reduce exposure to passive cigarette smoke	[12,19,155]

ESC/EAS: European Society of Cardiology/European Atherosclerosis Society, WHO: World Health Organization, TGs: triglycerides, TC: total cholesterol, BMI: body mass index, HDL-C: high-density lipoprotein cholesterol, LDL-C: low-density lipoprotein cholesterol, PUFA: polyunsaturated fatty acid, *—percent of total energy intake.

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
