# Peer review of "Non-Systematic Review of Diet and Nutritional Risk Factors of Cardiovascular Disease in Obesity"

_nutrients, 2020, doi:10.3390/nu12030814_

Round 1

Reviewer 1 Report

The present review discusses in a non-sistematic manner currently known diet and nutritional risk factors for CV disease in the setting of obesity.

Although very appealing and interesting, the manuscript needs an extensive revision. Some aspects have not been discussed, some others have been discussed in an incomplete way and must be enlarged. Take a look to remarks below.

Additionally, any figures or tables were provided. Tables and figures are very important to increase the understanding of the paper and are highly suggested.

Paragraph 2 - Obesity

  1. I would include what WHO states about obesity and CV risk in a more extensive way.
  2. With regard to possible mismatches of obesity definition based on BMI, please consider to discuss also these papers, which focus on this central aspect: for example sarcopenic obesity - Clin Nutr. 2019 Nov 27. pii: S0261-5614(19)33151-6. doi: 10.1016/j.clnu.2019.11.024; new proposed definitions - Obes Facts. 2017;10(4):284-307.
  3. Authors discussed about VAT and SAT, but some words about epicardial adipose tissue should be spent considering its importance with regard to AF, CAD, and HF. You can take a look to these papers: J Am Coll Cardiol. 2018 May 22;71(20):2360-2372; Heart Fail Rev. 2017 Nov;22(6):889-902; Compr Physiol. 2017 Jun 18;7(3):1051-1082. Additionally, what about pericardial adipose tissue? See Eur J Clin Invest. 2018 Jul;48(7):e12942; Eur Arch Psychiatry Clin Neurosci. 2018 Oct;268(7):719-725.

Paragraph 3 - Obesity paradox

I feel that this part can be part of the previous one. However, I'd suggest authors to enlarge a little bit the explanation of the obesity paradox, which is a little more complicated than what included here. In addition, the obesity paradox has been recently linked to aging. This may be of help for readers less familiar with this topic. Please consider to increase the knowledge about the obesity paradox, eventually by taking into consideration these papers: Prog Cardiovasc Dis. 2018 Jul - Aug;61(2):182-189; Prog Cardiovasc Dis. 2018 Jul - Aug;61(2):151-156; Vasc Health Risk Manag. 2019 May 1;15:89-100; Eur J Intern Med. 2018 Feb;48:6-17.

Paragraph 4 - Nutritional related CV risk factors

  1. This paragraph is not particularly sound. I found some mistakes. For example, why did authors state "The genesis and risk factors of atherosclerosis and oxidative stress are not fully characterized"? This is not true neither any reference was provided. Actually, we now know in great detail the pathogenesis of atherosclerosis. In particular, after the publication of the CANTOS study, we definitely now that atherosclerosis is an inflammatory disease. This must be corrected or deleted.
  2. This paragraph is overall confusing and needs to be reorganized. For example, a table or a figure can greatly help in this task. Authors must help readers to catch the key elements of the review.

Paragraph 5 - Appropriate diet in CVD prevention

  1. Main studies for each of the sub-paragraphs must be summarized in one or more tables to facilitate the reading. Please provide them. Please highlight in tables positive and negative aspects for each of the dietary patterns.
  2. While discussing about Mediterranean diet and DASH, please provide a figure summarizing main nutrients included in these diets discussing also the differences, if any. Alternatively, you can provide only a figure for the Mediterranean diet, while discussing differences with other diets within the manuscript.
  3. In this part, no citation has been provided with the PREDIMED study, which showed that a Mediterranean diet supplemented with extra-virgin olive oil or nuts reduced the incidence of major cardiovascular events in persons at high cardiovascular risk (N Engl J Med 2013; 368:1279-1290).
  4. Please summarize in a table main studies investigating the positive role played by the Mediterranean diet. This is highly informative for readers not familiar with the field.

Paragraph 6 - Current dietary recommendations

Again, this paragraph is difficult to read just as a list of recommendations. Please provide a table to summarize them. This is the easiest way to make the review more readable, otherwise a list of recommendations is hard to digest.

General comments

  1. All in all, the manuscript needs an extensive English editing. There are sentences really hard to read and many typos. Please ask an English mothertongue to read the manuscript, this will definitely improve it.
  2. Authors did not consider at all the impact of the diet on microbiota and how this may impact on CV risk. This is definitely an increasing field of research and a paragraph should be added to cover this aspect (see Curr Cardiol Rep. 2015 Dec;17(12):120).

Reviewer 2 Report

This review paper is very well-written and the information provided is of significance to nutrition and disease prevention. The author described several common diets that has been reported for cardiovascular prevention pertaining to the obese population. Overall, the manuscript is well organized, minor comments below.

Comments:

  1. It is suggested to provide the introduction starting from obesity and then relate it to CVD. Eg. Section #2 can be replaced by section #3.
  2. As paleo diet has been getting its attention for weight loss purpose, maybe the author can include a section for paleo diet and it’s adverse effect on CVD. Or at least include in the discussion or introduction for further information since studies have shown close relation among paleo diet, triethylamine-O-oxide, and CVD.

Round 2

Reviewer 1 Report

Authors addressed most of the reviewer's suggestions and comments.

The current manuscript looks like to be improved, but still some aspects have to be fixed.

- Please correct: Epidemiological studies have shown a relationship between obesity and cardiovascular morbidity, CVD mortality--> Epidemiological studies have shown a relationship between obesity and cardiovascular morbidity and CVD mortality

- I would maintain the clear separation for the 3 columns in Table 1, as provided in the Authors' reply, in order to facilitate the reading of the table.

- No word has been spent about the role of microbiota in diabetes, which is a CV disease. Please add a couple of sentences also on this.

- Section 7 may be renamed as Conclusion instead of Summary.

- Please check again for typos across the whole text.
